# Percutaneous Radiofrequency Ablation of Thyroid Carcinomas Ineligible for Surgery, in the Elderly

**Pierre Yves Marcy \*** , **Marc Tassart, Jean-Guillaume Marchand, Juliette Thariat, Alain Bizeau and Edouard Ghanassia**

PolyClinics ELSAN Group, Medipole Sud, Quartier Quiez, 83189 Ollioules, France; marc.tassart@aphp.fr (M.T.); jeanguillaume.marchand@gmail.com (J.-G.M.); jthariat@hotmail.com (J.T.); alain.bizeau@ch-toulon.fr (A.B.); drghanassia@gmail.com (E.G.)
\* Correspondence: brozpy@gmail.com

**Abstract:** Thirty to 50% of differentiated thyroid carcinomas include papillary thyroid microcarcinomas (mPTC). Most of these tumors remain clinically silent, have a bright prognosis and a disease-specific mortality <1%. Surgery has been recommended as first line-treatment by current guidelines, the standard treatment being lobectomy. However, surgery has some drawbacks, including potential recurrent laryngeal nerve paralysis, hypothyroidism, hypoparathyroidism, in -patient basis hospital stay, lifelong medication, scarring of the neck, and general anesthesia related risks. Moreover, elderly patients who present severe comorbidities, could be ineligible for surgery, and others may refuse invasive surgery. Another option supported by the American Thyroid Association is active surveillance. This option can be considered as unattractive and difficult to accept by European patients, as there is a 2–6% risk of disease progression. Percutaneous image-guided thermal ablation has been successfully applied in the treatment of liver and lung tumors in the 1990s and 2000s; and has recently been proposed as an alternative to surgery in patients presenting with thyroid diseases. This minimally invasive treatment has similar efficacy, fewer complications, better quality of life and cosmetic outcomes than surgery. We report herein two cases of radiofrequency ablation of mPTC and T2 PTC in elderly patients who were ineligible for surgery.

**Keywords:** thyroid carcinoma; papillary thyroid microcarcinoma; thyroid incidentaloma; active surveillance; thermal ablation; ultrasound; percutaneous radiofrequency; contrast enhanced ultrasound; PET-CT

## 1. Introduction

Papillary thyroid carcinoma (PTC) is the most frequent thyroid malignancy. Its increased incidence over the last few decades is attributable to the following factors; the extensive use and continual improvement of high frequency ultrasonography (US) of the neck since the 1990s [1], the establishment of precise ultrasound diagnostic criteria suspicious for malignancy [2] and Eu-TIRADS scoring [3], the wider use of real-time US guided fine needle aspiration cytology (FNAC) and Bethesda scoring [4]. PTC is mostly a well-differentiated slow-growing indolent tumor, with a disease specific mortality <1% [5,6]. Surgical lobectomy is commonly recommended, however, it might present some disadvantages including general anesthesia related risks, few-day hospital stay, potential iatrogenic hoarseness, hypoparathyroidism, hypothyroidism, lifelong medication, and neck cosmetic issues. Some patients can be ineligible for or may refuse thyroid surgery. Active surveillance (AS) has been adopted as a new option for low-risk papillary thyroid carcinoma. Drawbacks include patient's anxiety and risk of disease progression during US follow-up [7,8]. Recently, US-guided thermal ablation has shown excellent results as a third alternative treatment option for low-risk PTC [9]. We report herein two typical cases of percutaneous radiofrequency management of incidental PTC in the elderly [10].

## 2. Case Description

We present herein two cases of incidental evolving PTC in elderly patients who were not eligible for/or who refused surgery. Percutaneous radiofrequency ablation (RFA) was performed as there was no evidence of gross extrathyroidal extension and metastatic neck lymph nodes on pre operative US assessment.

### 2.1. Case 1

A 80 years old female patient, body mass index BMI of 31.5, with no particular medical or surgical history presented with a central left 7 mm evolving (+50% volume increase within three months) thyroid nodule Eu-TIRADS5, Bethesda B6, that was incidentally depicted during a systematic Doppler arterial US examination in 2022 January. There was no suspicious lymph node, and the patient refused both surgery and active surveillance.

Multidisciplinary decision was taken to treat this evolving T1a N0M0 mPTC in 2022 May by using percutaneous RFA thermal ablation and US follow-up at 3, 6, 9 and 13 months (Figure 1).

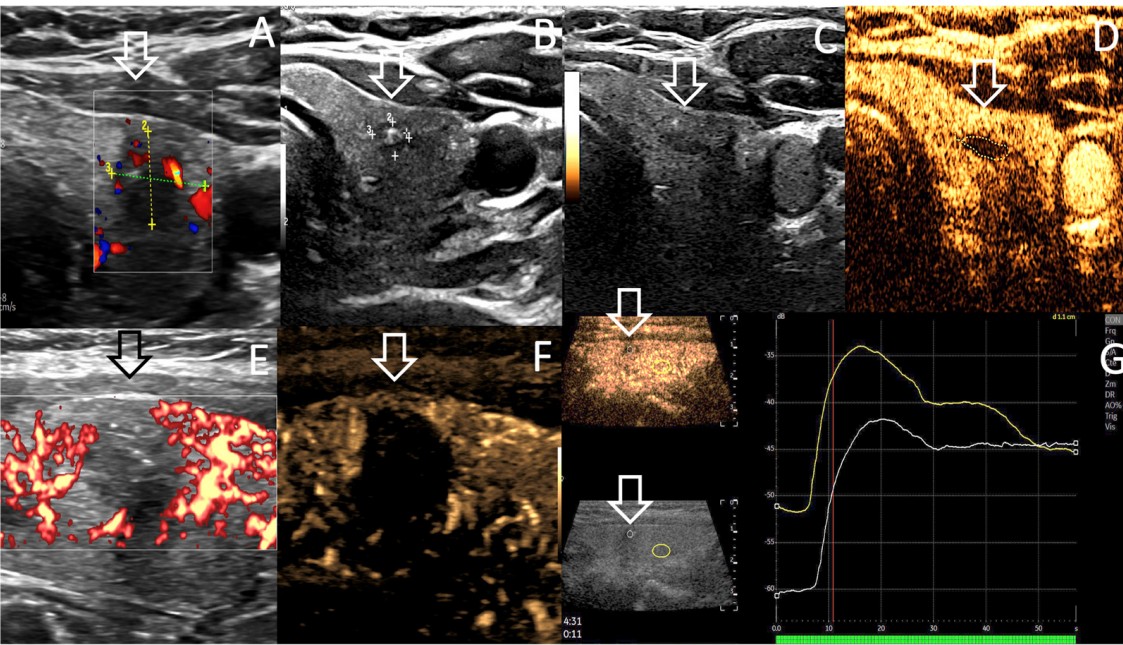

**Figure 1.** Case 1. US features showing 93% of volume reduction rate (VRR) of the treated central T1a mPTC of left thyroid, over 13 month time period; (**A**). Volume 0.4 mL before RFA; (**B**). Volume < 0.1 mL at 13 month follow up. (**C,D**). US scan without (**C**) and with contrast enhanced ultrasound CEUS (**D**) at 13 months display a 0.04 mL hypovascular area. (**E,F**). Power Doppler sagittal B-mode (**E**) and B flow (microvascular imaging) (**F**) US scanning at RFA completion display avascular treated nodule (arrow). CEUS show on axial (**D**) and sagittal views (**G**) the avascular circumscribed treated area (arrow) and the difference of vascularity at the arterial phase (12 s after CEUS) between normal adjacent thyroid tissue (yellow curve) and RFA treated nodule area (white curve, arrow). FNAC was performed both at the central and peripheral areas, depicting no suspicious cell at 13 months follow-up.

A 18 G single VIVA internally cooled electrode needle with 5 mm active tip (STARmed Co, Goyang-si, Gyeonggi-do, Republic of Korea) was used to deliver thermal ablation. Global delivered energy was 1171.5 Joules representing approximately 2240 J/mL of the treated nodule. Mean procedure pain assessed by numerical rating scale was 0.5/10 and post procedure related pain was nil [11]. Patient's prescription included prednisolone 60 mg during three days, 40 mg at day 4, 20 mg at day 5, and 10 mg at day 6; in association with omeprazole 20 mg/six days.

### 2.2. Case 2

A 88 years old metastatic prostate adenocarcinoma patient presented high uptake left thyroid focus (SUV max = 11.7 versus 4.2 in 2013) on 18-F fluorocholine PET-CT during oncology follow-up in June 2021 (Figure 2). Patient had undergone prostatectomy in 2000, and experienced right common iliac lymphadenopathy recurrence in 2013 successfully treated by hormonotherapy which was stopped in 2014. Post operative PSA serum level was 0.01 in 2014, increased up to 11.7 in May 2020, 17.9 in December 2020, and then decreased (0.14) and normalized (<0.01) after Triptoreline therapy had been initiated in May 2021. US depicted a vascularized solitary 19.8 × 18.4 × 22.2 mm = 4.2 mL Eu-TIRADS 4 nodule located at the inferior left thyroid, 3 mm distant to the thyroid capsule, with no suspicious lymph nodes (Figure 2B). FNAC revealed a Bethesda BVI, BRAF V600 mutated, T1b oxyphilic variant PTC. As patient comorbidities included severe persistent asthma since childhood, centrolobular emphysema, distal constrictive bronchiolitis, pulmonary and cardiac insufficiency with worsening dyspnea on exertion, cardiac arrhythmia (atrial fibrillation), patient underwent ultrasonography active surveillance (AS). Significant volume progression of the T1b PTC was noted from 4.2 mL in June 2021 to 7.6 mL, (23 × 21.3 × 29.6 mm), April 2022, without lymph node recurrence. There was no subcapsular tumor spread. Multidisciplinary decision was taken to treat this evolving T2 N0M0 malignant thyroid nodule by using percutaneous RFA thermal ablation and US follow-up at 3, 6, 9 and 12 months. Patient 's medical history included purulent pleuritis, drained spontaneous pneumothorax, duodenal ulcer and hiatal hernia. Taking into account medical history and comorbidities, heavy sedation was contraindicated by the anesthetist. Patient underwent percutaneous RFA under local anesthesia during an ambulatory procedure in June 2022. Paracetamol 1 g intra venous (iv), and ketoprofene 100 mg iv were given at procedure. A 18G internally cooled needle with 7 mm active tip, (VO Medica, Guyancourt, France) was used to perform the procedure. Global delivered energy was 27,740 J representing approximately 3500 J/mL of the treated nodule. Mean procedure pain assessed by numerical rating scale was 3/10 and post procedure related pain was nil.

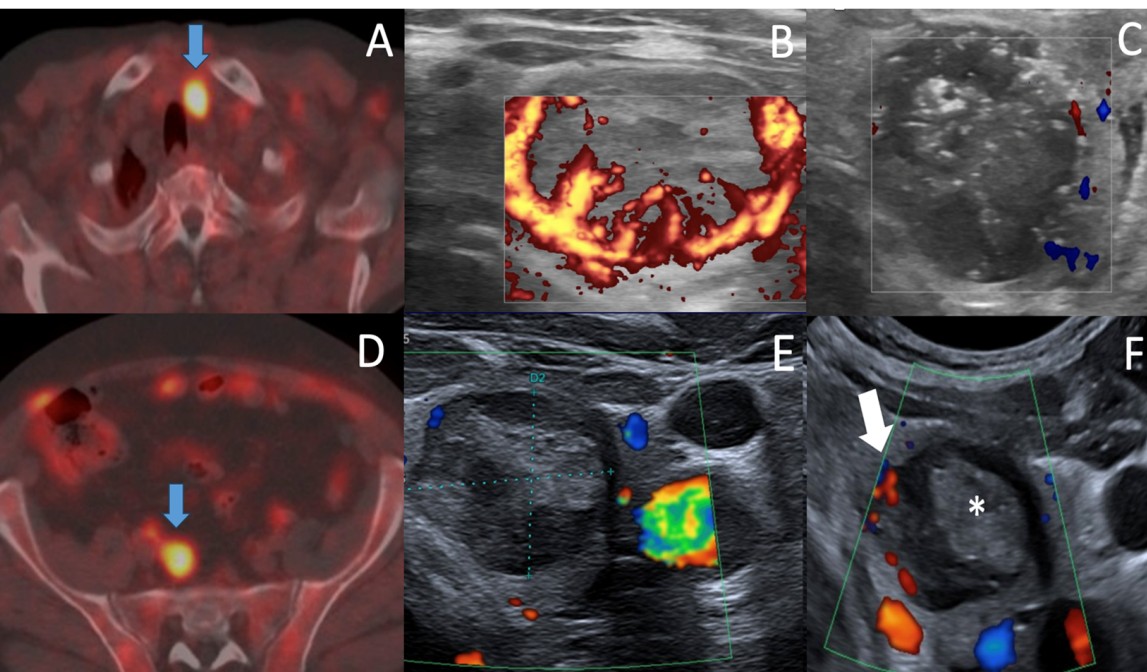

**Figure 2.** Case 2. $^{18}$F Fluorocholine PET-CT (**A**,**D**) showing high focal uptake on left thyroid ((**A**), arrow) SUV max = 11.7, June 2021 (versus 4.2 in 2013) and right common iliac lymph node uptake ((**D**), arrow) corresponding to prostate cancer recurrence. Triptoreline therapy was thus initiated. Axial US assessment

before and after RFA procedure, June 2022 (**B**,**C**) shows Eu-TIRADS5 Bethesda B5, hypervascular oxyphilic thyroid T2 N0 PTC, 23 × 21.3 × 29.6 mm = 7.8 mL (**B**) and the slight increased in size of the treated nodule at RFA completion = 8.9 mL due to intranodular oedema and safety margin RFA. Note the appearance of echogenic foci (**C**) due to the presence of air, hemorrhage foci related to thermal ablation. Axial US follow-up (**E**,**F**) showing break of tumor regression curve after 6 months ((**E**) 2.2 mL) & vascularized regrowth in the medial part of ablation area at ten months (Arrow (**F**), 2.2 mL). (**F**). FNAC was performed at the central (*) and medial (arrow) parts of the ablation area, and depicted B5 local regrowth in a 0.3 mL hypoechoic hypervascular medial part (arrow) of the ablation area at ten months of follow-up.

### 2.3. Technical Aspects of RFA Procedure

Both RFA procedures were performed by expert radiologists (PYM, MT; more than 30 years of percutaneous interventional radiology experience, more than six years (PYM) and three years (MT) of thyroid RFA experience) on an ambulatory outpatient basis, under sterile conditions, local anesthesia, using US real time monitoring. Patients were lying in supine position with neck extension, in the interventional radiology suite. Patients were asked to fast for three hours before procedure. An 18-gauge thyroid-dedicated internally cooled electrode with 0.5/0.7 cm active tips were used. The relationships between the tumor and critical functional structures such as vagus nerve, middle sympathetic cervical ganglion, esophagus, and trachea, normal parathyroid glands were carefully evaluated to prevent injury (Figure 3). Twenty ml of 2% lidocaine solution were mixed with 10 mL HCO3Na solution for local anesthesia at the puncture site and peri-thyroidal area in both cases. A trans-isthmic approach was used to prevent damage to the danger zone (Figure 3). The moving shot technique [12] was performed, using an initial RF power ranging from 25 to 40 W. If a transient hyperechoic zone did not form at the electrode tip within 5–10 s during treatment, the RF power was increased in 5 to 10 W increments, up to 40 W.

A 22-gauge × 4 cm needle was used to inject a 20 mL cooled dextrose solution between the tumor and critical functional structures to prevent thermal injury, the so-called hydrodissection technique, in patient 1 [13]. To prevent marginal recurrence, a 2 mm thickness of a perinodular normal thyroid tissue was also ablated ("safety margin procedure"). Patients were carefully observed for 3 h in the hospital after ablation, and left hospital after having had a light snack; and a final clinical and neck US assessment in patient 1, clinical assessment in patient 2, by the interventional radiologist. Patients were phone called at day 1 and day 2.

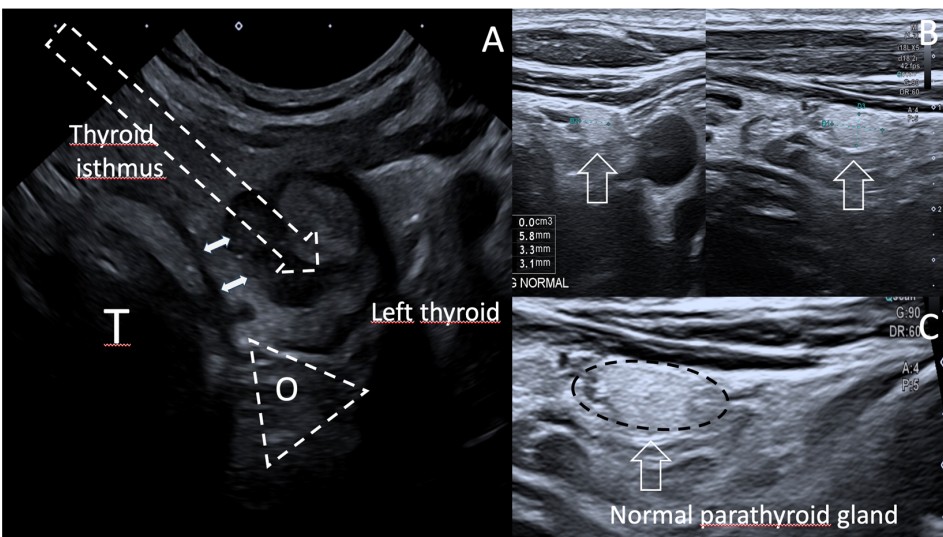

**Figure 3.** (**A**). Axial US scan of the Danger Zone, Case 2, at ten months. The Oesophageal tracheal groove (TEG) is a triangular danger zone where recurrent laryngeal nerve (RLN) can be injured, as

well as oesophagus (O) and tracheal wall (T). This is prevented by performing an oblique in-plane US-guided trans isthmic approach of the electrode (arrow). Double arrows show a 2.2 mm thickness of normal thyroid tissue between tracheal wall and local recurrence nodule. (**B**,**C**). US diagnosis of normal parathyroid gland. US axial & sagittal scans below the lower posterior left thyroid easily depict an ovoid, homogeneous, markedly hyperechoic 5.8 × 3.3 × 3.1 mm lower left parathyroid gland, typically brighter than the thyroid gland [14].

### 2.4. Follow-Up and Results

The volume of tumor was calculated as V = πabc/6 (where V is the volume, a is the largest diameter, and b and c are the two other perpendicular maximal diameters) and reported at each US follow up examination [6,15]. Both patients were assessed by using color Doppler, power Doppler US, and microvascular imaging at one month, three months, six months, nine/ten months and twelve/thirteen months in cases 1 & 2 respectively. CEUS was performed in case 1 at thirteen months post procedure. Both patients underwent FNAC systematically at 13 months (patient 1), and in case of suspicion of tumor local regrowth, earlier in case 2, at ten months.

There was significant constant volume reduction until the thirteenth month in case 1, and until the sixth month in case 2 (Table 1).

**Table 1.** Characteristics of PTC RFA procedures and follow-up in cases 1 & 2.

| Work Up, RFA & Follow-Up | Case 1<br>80 Years Old, Female<br><br>T1aN0, Eu-TIRADS5, B6 | | Case 2<br>88 Years Old, Male<br>Oligometastatic PROSTATE<br>T2N0 PTC, Eu-TIRADS5, B5 |
|---|---|---|---|
| Active surveillance | No | | 12 months (2021–2022, June) |
| Progression | 50% volume before RFA T1a < 10 mm | | (T1b to T2) shift |
| Thyroid volume: right + left lobes | 6 + 7 = 13 mL | | 6 + 15 = 21 mL |
| TSH | Normal | | Normal |
| Calcitonin | Normal | | Normal |
| RFA procedure | May 2022 | | June 2022 |
| Delivered Energy | 2240 J/mL | | 3500 J/mL |
| Initial volume | 0.4 mL | | 7.8 mL |
| Pain assessment | | | |
| Per RFA | 0.5/10 | | 3/10 |
| Post RFA | 0/10 | | 0/10 |
| Post–RFA Follow-up | RFA | 0.7 | 8.9 |
| Volume assessment (mL) | 1 month | 0.42 | 3.4 |
| | 3 months | 0.1 | 2.7 |
| | 6 months | 0.05 | 2.2 |
| | 9/10 months | 0.03 | 2.2 |
| | 13/12 months | <0.03 | 2.2 |
| VRR (%) | VRR | 93% | 75.3% |
| Regrowth depiction in ablation area | None (13 months)<br><br>Color Doppler/MVI/CEUS | | Medial part (0.3 mL, at 10 months)<br><br>Color Doppler/MVI |

Both tumors showed significant constant volume decrease in case 1 (93% VRR at 13 months) and 75.3% VRR at six months in case 2. No local regrowth was depicted in case 1, whereas local medial regrowth was depicted in the medial aspect of the ablation area (Figure 2F in case 2), thanks to Color Doppler assessment, at ten months. Initial volumes of PTC were respectively 0.4 mL and 7.8 mL before RFA procedure, and post-RFA treated nodule volumes were 0.7 mL and 8.9 mL respectively, due to safety margin appliance. There is a break of the VRR curve after 6 months follow-up in case 2.

At follow-up, thyroid function tests remained normal, and no complication was recorded at one week, one month, three, six nine and twelve/thirteen months.

The volume reduction rate was calculated as follows: volume reduction ratio (VRR) = ([initial volume − final volume] × 100)/initial volume [6].

There was no recurrence in case 1, the VRR was estimated at 93%. Doppler US, CEUS examination and FNAC did not reveal local recurrence in case 1 whereas VRR in case 2 was 75.3% with a break of the curve after the sixth month. Local B5 malignant regrowth was proven on FNAC of medial part at ten months (Figure 2F, arrow).

## 3. Discussion

In 1993, seven years after the Tchernobyl nuclear accident, Furmanchuk et al. reported an autopsy malignancy rate of 9.3% including mostly occult mPTC (8.8%), and one medullary thyroid carcinoma. There was no significant mPTC sex ratio between male and female patients, less than one third was multifocal; mean diameter was less than 5 mm [16]. In 1997, Pacini et al. reported that post-Chernobyl Belarus thyroid carcinomas exhibited greater aggressiveness at presentation, affected younger patients, with increased thyroid autoimmunity, when compared with naturally occurring thyroid carcinomas of the same age group observed in Italy and France [17]. Recently, Robenshtok et al. reported a malignancy rate of 8.5% in a meta analysis of 29 autopsy studies regrouping 8750 patients without known history of thyroid cancer. Interestingly, multifocality rate was reported in 28.2% of cases, bilateral involvement in 18%, minimal extra thyroid extension (ETE) in 24.5%, presence of lymph node (LN) metastases in 11%, vascular invasion in 16%, and there was 0.004% of distant visceral metastases [18]. Thus the authors concluded that, distant metastases excluded, occult mPTC multifocality, bilateral involvement, ETE, LN metastases, vascular invasion may no longer be considered as signs of aggressive disease and an indication for completion thyroidectomy or radioiodine therapy. Moreover, when considering low-risk PTC < 20 mm longest diameter Canadian patients who were asked for choosing between surgery and active surveillance (AS), majority of concerned patients (77.5%) opted for AS. Factors associated with choosing AS included older age, lower education level, and having a surgeon outside the study institution [19].

Interestingly, Cho et al., in a Korean metaanalysis [20], reported a very low pooled proportion (5.3%) of tumor size enlargement occurring at 5 years, and a 1.6% five-year lymph node metastasis during active surveillance. Given the evidence of overdiagnosis and overtreatment of PTC, the 2015 American Thyroid Association adult guidelines recommended lobectomy versus total thyroidectomy and a restricted use of postsurgical radioactive iodine in the management of PTC presenting with low recurrence risk [21]. Active surveillance and non surgical management are the other two options.

In the 1990s, Livraghi et al. reported satisfactory results of image-guided thermal ablation of hepatocellular carcinomas not eligible for surgery [22]; a decade later on, during the 2000s, de Baere et al. reported eradication of 91% of liver metastases that were not suitable for surgery [23]. In Italy, Papini et al. were the first authors to report a successful percutaneous laser ablation of incidental mPTC in a patient non eligible for surgery [24], Valcalvi et al. reported three other cases in 2013 [25]. Regarding the best management of indolent PTC, percutaneous image-guided thermal ablation studies have shown to provide a curative treatment, on an outpatient basis, without heavy sedation, minimizing the invasiveness of treatment itself, and preserving the thyroid and parathyroid gland function [26–29]. Some series have been reported, mostly from Asian Chinese and Korean authors, on the application of real-time US-guided thermal ablations in the treatment of mPTC, also showing favorable results [30–32]. Recently, Mauri et al. reported their encouraging preliminary results regarding US-guided RFA of mPTC in eight patients who were given the choice of three alternatives at a multidisciplinary council, as follows: image-guided RFA, surgery, and active surveillance. Technical success rate was 100%. Ablated volume significantly reduced from $0.87 \pm 0.67$ mL at first follow-up to $0.17 \pm 0.36$ at last follow-up ($p = 0.003$) [26]. There was no change in thyroid function tests, and no minor or major complications were recorded at a minimum follow-up of 10.2 months. Patient's satisfaction was high (10/10), mean pain during ablation was $1.4/10 \pm 1.7$, and mean

pain after RFA procedure was 1.2/10 ± 1.1. Cho et al. reported the high and persistent accuracy of mPTC RFA of 84mPTC nodules in 74 patients after more than five years of follow up (72 months). Complete disappearance rates of 98.8% and 100% were achieved at 24 and 60-month follow-up respectively, with a mean number of RFA sessions of 1.2. Additional ablations were performed in 13 of 84 tumors. The four newly developed cancers in three patients were also treated with RFA and completely shrank. During the follow-up period, there was no local tumor progression, no LN or distant metastasis, and no patients underwent delayed neck surgery. The major complication rate was 1.4%, and there was no delayed complication or procedure-related death [20].

During a mean follow-up time of 49.25 ± 12.98 months, Yan et al. reported on a VRR of 99.40 ± 4.43% and an overall incidence of local tumor progression of 3.7% (18/487). The authors looked for the impact of multifocality on the efficacy of treatment and recurrence. The comparison between the multifocal group and unifocal group did not show significant differences, as follows: complete disappearance rate was 95.61% vs. 89.09%, there was similar local tumor progression 5.45%, LN metastasis rate was 1.82% vs. 0%, recurrent mPTC rate 1.82% vs. 5.45%, persistent lesions 1.82% vs. 0%, and recurrence free survival (RFS) rate 96.36% vs. 94.55% respectively. No distant metastasis or delayed surgery were reported at follow-up [30].

Xue et al. reported on, in a recent metaanalysis pooling ten non-randomized controlled trials and 1279 pooled patients that percutaneous RFA treatment significantly reduced the volume reduction ratio (VRR) at 12 months of 93.27% and that the complete disappearance rate was 64% at 12 months post RFA. Additionally, pooled results showed the incidence of mPTC residue in ablation area, newly discovered mPTC and LN metastases after RFA treatment were respectively 0.3%, 2.5% and 1.0%; and the incidence of complications after RFA treatment was 1.8% in the 753 treated mPTC patients, mostly including pain (0.3%) and hoarseness (0.6%) [33].

The second metaanalysis by van Dijk et al. pooled fifteen studies and a total of 1770 patients, 1379 females [77.9%], mean age, 45.4 years (range, 42.5–66.0 years) with 1822 tumors treated with RFA. Fourty-nine patients out of 1822 (=2.7%) underwent one additional RFA session and 1 tumor underwent 2 additional RFA sessions [34]. The pooled complete disappearance rate at the end of follow-up was 79% at 33 months follow-up. VRR at 12 months after RFA was 92.1%. The overall tumor progression rate was 1.5%, local residual mPTC in the ablation area was 0.4%, newly discovered mPTC was 0.9%, and four patients (=0.2%) of the 1770 treated patients developed lymph node metastases during follow-up. No distant metastases were detected. The pooled proportion of total complication rate was 2%.

In our present cases, patients were informed about advantages and limitations of percutaneous RFA in comparison with the other management options. RFA decision was made in accordance with the patient, by referent physicians at a multidisciplinary council, taking into account the tumor progression of incidentally diagnosed PTC, the refusal for surgery and willingness to undergo safe quick efficient curative treatment with preservation of the thyroid gland in case 1, the ineligibility of surgery with strong tumor progression in severe comorbidity patient in case 2 after one year of active surveillance [35,36]. As a 2 mm margin was required to technically perform the procedure in safe conditions, a trade-off had to be found rapidly between active surveillance of an "indolent growing lesion" and passing up the unique opportunity of safely performing percutaneous local treatment. As a matter of fact, percutaneous RFA was performed according to the guidelines edicted by Mauri et al. and Papini et al. in case 1, as a curative safe and effective treatment of mPTC [37,38].

Case 2 was more debatable as there is currently no guidelines, which may formally valid RFA in larger T2 thyroid tumors, even if RFA indication of mPTC T1a may extend to T1b thyroid tumors. Interestingly Zhao et al. reported the complication rate (recurrent laryngeal nerve RLN palsy) of RFA to be related to the size and malignant cause of the thyroid nodule as follows: 2.9% in benign thyroid nodules, and 6.3% in PTC, 5% in T1a

PTC, 10.7% in T1b PTC; and 28.6% in T2 PTC patients, as in case 2 [39]. In the PTC group, the tracheoesophageal groove (TEG) distance, the anterior capsule distance, the median capsule distance, the posterior capsule distance, and maximum nodule diameter were risk factors for RLN injury. Furthermore, the authors advocated for a minimally isolating fluid continuously injected between tracheal wall and medial thyroid lobe to prevent RLN injury and also tracheal/oesophageal injury [40]. They recommend performing hydrodissection of the danger zone of at a minimum of 3.9 mm thickness. Taking into account these considerations, as T2N0 RFA technique is still a work-in-progress procedure, we plan to perform a large hydrodissection at the danger triangle (Figure 3A) to retreat the local regrowth in the ablation area (Figure 2F), safely and efficiently in the near future, on patient 2's request, and after multidisciplinary council advice.

Interestingly, it is worthy to note that as RLN is anatomically running more vertical on the left side of the TEG compared to the right side, the RFA retreatment procedure appears to be a little less risky than that on the right side. Moreover, there was still a 2.2 mm non tumoral thyroid tissue between the TEG and the left PTC medial regrowth on last US scanning of the neck (Figure 3A), thus giving the opportunity to safely retreat this medial part of the nodule.

It is also important to keep in mind that the RLN palsy incidence after RFA among 1004 RFA treated patients was 6.3%, to be compared to the 6.0% rate reported in 11,370 surgery patients in Gunn et al.'s study [41].

As parathyroid gland preservation is also a main thyroid surgery issue, we would therefore point out that neck US may frequently identify even normal sized parathyroid glands at the posterior aspect of the lower thyroid (Figure 3B), whose US features easily allow to differentiate from metastatic recurrent lymph nodes of level VI 14].

The results of our preliminary experience show that image-guided thermal ablation can be safely applied in the treatment of small indolent PTC, offering a potentially curative, minimally invasive treatment to patients in alternative to surgical resection or active surveillance (Figure 4). This highlights the critical relevance of an accurate US evaluation before during and after RFA treatment, as well as proper selection of dedicated patients. Accurate US follow-up by expert radiologists is also of utmost importance to depict and further retreat local recurrence.

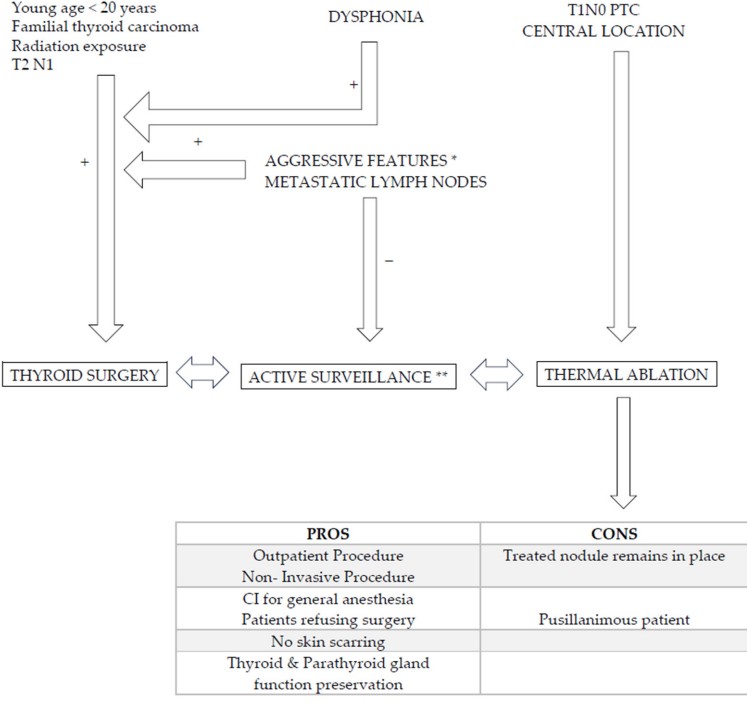

**Figure 4.** Algorithm regarding PTC management and alternative options to surgery. Young age, familial

primary thyroid carcinoma context, previous radiation therapy of neck, RLN palsy (dysphonia), tumor located in the isthmus, at the oesotracheal triangle and/or close to tracheal wall, extra thyroid extension (R+), presence of metastatic neck nodes, aggressive behavior * (cytology, molecular analysis) on FNA should lead to surgery [9]. Active surveillance ** (AS) is an alternative to surgery as well as RFA. AS is performed every six months/2 years: volume increase > 50%, long axis increase > 3 mm and/or appearance of neck lymph node should lead to surgery. RFA is typical indicated in central located T1a,b N0 monofocal lesion in patients refusing or contrain-dicated for surgery and refusing AS. RFA pros and cons are reported.

## 4. Conclusions

These two case reports including (us) T1a and (us) T2 PTC in the elderly were technically successfully treated with percutaneous RFA with no related complications. They illustrate the actual situation where indolent T1 PTC may benefit from active surveillance and/or RFA treatment. They support the evolving leading position of neck high frequency ultrasound in the decision making algorithm and treatment of small low risk papillary thyroid carcinomas, namely in the elderly. RFA provides an alternative and intermediate solution between lobectomy (T1a, b, and T2) and active surveillance (AS) and could be considered in the near future as a step-up minimally invasive treatment of local PTC growth under active surveillance or as an initial treatment in PTC patients being anxious about AS and/or refusing or being not eligible for surgery. Further prospective studies are nevertheless needed to validate the role of RFA in (us) T2 N0 PTC patients who are ineligible or contraindicated for thyroid surgery.

**Author Contributions:** Conceptualization, P.Y.M., J.-G.M., J.T. and E.G.; Methodology, M.T. and A.B.; software, A.B.; validation, P.Y.M. and M.T.; formal analysis, M.T. and P.Y.M.; investigation, P.Y.M. and M.T.; resources, P.Y.M. and M.T.; data curation, P.Y.M. and M.T.; writing—original draft preparation, P.Y.M.; writing—review and editing, P.Y.M. and A.B.; visualization, M.T.; supervision, M.T.; project administration, P.Y.M., M.T. and A.B. All authors have read and agreed to the published version of the manuscript.

**Funding:** This research received no external funding.

**Institutional Review Board Statement:** These are case reports; decision for RFA of patient 1 was given by Dr AB, referring head and neck surgeon, and pluridisciplinary council, after written informed consent of the two patients. The study was conducted in accordance with the Declaration of Helsinki, and approval by the Ethics Committee of PolyClinics ELSAN Medipole Sud was waived as this is a case report.

**Informed Consent Statement:** Written Informed consent was obtained from the two subjects involved in the study.

**Data Availability Statement:** Not applicable.

**Acknowledgments:** This publication is performed on behalf of the AFTHY, Association Française de Thyroïdologie.

**Conflicts of Interest:** All the authors declare no conflict of interest.

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
