# Peer review of "Percutaneous Radiofrequency Ablation of Thyroid Carcinomas Ineligible for Surgery, in the Elderly"

_curroncol, doi:10.3390/curroncol30080539_

Round 1
Reviewer 1 Report
The manuscript describes two cases of radiofrequency ablation of thyroid cancer as an alternative to thyroid surgery. The concept is not novel, and a significant volume of literature is available on the subject. However, the cases are well described, and the discussion of the existing evidence made by the authors is valuable. I have minor comments.
Lines 35-38. Hypoparathyroidism is an unlikely complication of lobectomy. Also, at our institution, lobectomy is an outpatient procedure, therefore prolonged (few days) hospital stay is also rare.
Figure 3 B,C. Lines 297-300. Normal parathyroid glands are usually not seen on ultrasound. Therefore, direct visualization is unreliable in preventing parathyroid damage during RFA. However, even if a single parathyroid gland is affected during the procedure, hypoparathyroidism is very unlikely. My understanding also, is that parathyroid glands are usually hypoechoic on ultrasound.
Author Response
Dear Reviewer,
Thank you for your comment.
I have minor comments.
Lines 35-38. Hypoparathyroidism is an unlikely complication of lobectomy. Also, at our institution, lobectomy is an outpatient procedure, therefore prolonged (few days) hospital stay is also rare.
Thank you for this comment. I did not know that thyroid lobectomy could be performed on an outpatient basis.
Figure 3 B,C. Lines 297-300. Normal parathyroid glands are usually not seen on ultrasound. Therefore, direct visualization is unreliable in preventing parathyroid damage during RFA. However, even if a single parathyroid gland is affected during the procedure, hypoparathyroidism is very unlikely. My understanding also, is that parathyroid glands are usually hypoechoic on ultrasound.
Thank you for your comments. the following reference clearly states that the normal parathyroid gland is currently depicted in almost 75% of normal patients, thanks to the high frequency US material, as shown on figure 3.
The normal parathyroid gland appears as "hyperechoic tissue" as shown in the following reference: US features of normal parathyroid glands: a comparison with metastatic lymph nodes in thyroid cancer; Seong Ju Kim', Dong Gyu Na', Byeong-Joo Noh, Ultrasonography 2023; 42(2): 203-213. https://doi.org/10.14366/usg.22119
Reviewer 2 Report
Since the explanation of "TIRAD" does not appear, please put it where it appears first.
Figure 1
The letters in the figures are so small that it is difficult to tell briefly which figure they are referring to. It is better to use a slightly larger font size.
At least, I think it is better for letter to be the same as or larger than the text size of the article.
Line 67
13month → 13 months
Line 77, Line 103 (Figure 2)
18- F → 18F is better.
Line 82
Why is Triptoreline shaded?
Line 98
“iv” mean “intra venous injection”???
Table 2
I think that it is easier to see if the font size of Table2 is lowered a little.
Line 198-201
These sentences are completely the same as reference 20.
In reference section “20.x”, what is “.x”?
Line 222
Why are you displaying markers?
Line 227-231
These sentences are completely the same as reference 32.
Shouldn't it be "quote" instead of "reference"?
Since it is written as Case 1 and Case 2 in methods, I think that Case 1 or 2 is easier to understand than patient 1 or 2 in Table 1, but what do you think? It may be redundant, but it is easier to understand if the correspondence between Case and Patient is described.
Line 250
patients (1379 female [77.9%]; mean age, 45.4 years (range, 42.5-66.0 years)
Is this ”(” required?
The numbers in Table 1 are marked with an asterisk but are not explained in footnotes.
Although "cosmetic outcome" is written in the abstract, in the other hand, the presented data of the patients are internal body rather than surface.
Since it's called "cosmetic", wouldn't it have been easier to understand if there was a photograph of the outer skin, etc., if possible, showing that the damage was minimal?
I think it is a case report with few examples, but I think it will be easier to understand if there is a diagram (something like a chart diagram) that can explain the superiority of the treatment method selected this time.
Author Response
Dear Reviewer, Thank you for your comments, the modifications have been made in italics.
Figure 1
The letters in the figures are so small that it is difficult to tell briefly which figure they are referring to. It is better to use a slightly larger font size.
Corrections have been made accordingly.
Line 67
13month → 13 months
OK
Line 77, Line 103 (Figure 2)
18- F → 18F is better.
OK
Line 82
Why is Triptoreline shaded?
This has been corrected, this was a typo.
Line 98
“iv” mean “intra venous injection”???
IV means intravenous and has been modified.
Table 2
I think that it is easier to see if the font size of Table2 is lowered a little.
Thank you, corrections have been made from Palatino font size 10 to 9.
Line 198-201
These sentences are completely the same as reference 20.
Thank you, the sentence has been modified accordingly.
In reference section “20.x”, what is “.x”?
X is a typo and has been corrected.
Line 222
Why are you displaying markers?
We have modified : "signs" instead of "markers". Thank you.
Line 227-231
These sentences are completely the same as reference 32.
Shouldn't it be "quote" instead of "reference"?
Thank you, the sentence has been modified.
Since it is written as Case 1 and Case 2 in methods, I think that Case 1 or 2 is easier to understand than patient 1 or 2 in Table 1, but what do you think? It may be redundant, but it is easier to understand if the correspondence between Case and Patient is described.
OK, we have made the modifications in Table 1, and further on in the manuscript.
Line 250
patients (1379 female [77.9%]; mean age, 45.4 years (range, 42.5-66.0 years)
Is this ”(” required?
Modifications have been made.
The numbers in Table 1 are marked with an asterisk but are not explained in footnotes.
Asterisk has been deleted.
Although "cosmetic outcome" is written in the abstract, in the other hand, the presented data of the patients are internal body rather than surface.
Since it's called "cosmetic", wouldn't it have been easier to understand if there was a photograph of the outer skin, etc., if possible, showing that the damage was minimal?
Thank you for your remark, there is no scar at all after RFA, and we consider that it is not something worth to display a normal neck before and after RFA .
I think it is a case report with few examples, but I think it will be easier to understand if there is a diagram (something like a chart diagram) that can explain the superiority of the treatment method selected this time.
Thank you, we have added a chart diagram.